# Genome-Wide Identification and Functional Prediction of LRR-RLK Family Genes in Foxtail Millet (*Setaria italica*) in Response to Stress

**DOI:** 10.3390/ijms26104576

**Published:** 2025-05-10

**Authors:** Zhijiang Li, Xinmiao Kang, Miaomiao Song, Xiaojie Dong, Jinfeng Ma, Jinhai Yu, Xiangyu Li, Yalu Zheng, Guangquan Sun, Xianmin Diao, Xiaotong Liu

**Affiliations:** 1Heilongjiang Provincial Key Laboratory for Genetic Improvement of Minor Grain Crops, Institute of Crop Resources, Heilongjiang Academy of Agricultural Sciences, Harbin 150086, China; lizhijiang12@163.com (Z.L.); dongxiaojie0605@163.com (X.D.); hljmjf@163.com (J.M.); hnygy@163.com (J.Y.); xiangyu527443@aliyun.com (X.L.); zhengyalu001@163.com (Y.Z.); sunguangquan@163.com (G.S.); 2Key Laboratory of Molecular and Cellular Biology of Ministry of Education, Hebei Research Center of the Basic Discipline of Cell Biology, Hebei Collaboration Innovation Center for Cell Signaling, Hebei Key Laboratory of Molecular and Cellular Biology, College of Life Sciences, Hebei Normal University, Shijiazhuang 050024, China; kangmiaomiao1314@163.com (X.K.); 18232891057@163.com (M.S.); 3Institute of Crop Sciences, Chinese Academy of Agricultural Sciences, Beijing 100081, China

**Keywords:** LRR-RLK, foxtail millet, evolution analysis, stress response

## Abstract

Leucine-rich repeat receptor-like kinases (LRR-RLKs) are involved in the regulation of various biological processes, including plant growth, development, and responses to biotic and abiotic stresses. Foxtail millet (*Setaria italica*), an important cereal crop, has been extensively studied for its stress tolerance mechanisms. In this study, we performed a comprehensive phylogenetic analysis and chromosomal mapping of LRR-RLK genes in *Setaria italica*. A total of 285 *SiLRR-RLK* genes were identified and classified into 12 subfamilies based on phylogenetic relationships. Chromosome localization analysis revealed that *SiLRR-RLK* genes are unevenly distributed across the chromosomes, with certain regions showing gene clusters. Functional analysis of these genes under biotic and abiotic stress conditions suggested that several *SiLRR-RLK* family members are involved in key stress response pathways. Expression profiling indicated differential expression patterns of *SiLRR-RLK* genes in response to various stresses, including drought, salinity, and pathogen infection, highlighting their potential roles in stress adaptation. In conclusion, the phylogenetic and functional analysis of the *SiLRR-RLK* gene family in *Setaria italica* provides valuable insights into their roles in stress responses and lays the groundwork for future studies aimed at enhancing stress tolerance in foxtail millet.

## 1. Introduction

In plant cells, the transmission of signals from the external environment to the interior requires the perception of surface receptors. The leucine-rich repeat receptor-like kinase (LRR-RLK) family represents a critical class of membrane receptor proteins in plants [1]. These proteins function as “sensors” on the cell membrane, regulating essential physiological processes such as plant growth and development, immune responses, and environmental adaptations by sensing external signals [2,3,4,5,6,7,8].

The LRR-RLK family is one of the largest and most diverse receptor kinase families in plants, primarily composed of an extracellular domain (ECD) that senses signals, a transmembrane domain that anchors the protein within the membrane, and an intracellular kinase domain (KD) that transduces signals to downstream pathways through autophosphorylation. The ECD is the core component of LRR-RLKs, typically consisting of multiple leucine (Leu) residues and other alternating amino acids, forming a unique “horseshoe” structure. This conformation facilitates receptor–ligand binding and the recognition of external signals. For instance, the LRR structure of the FLS2 receptor specifically recognizes the bacterial flagellin peptide (flg22), subsequently activating the plant’s immune response [3]. The LRR structure of the BRI1 protein specifically recognizes brassinolide (BL), subsequently activating the brassinosteroid signaling pathway, which regulates various growth and developmental processes in plants [9,10]. The number and arrangement of LRRs vary among different LRR-RLK family members, providing a basis for their functional diversity [11].

LRR-RLKs contain one or more transmembrane domains that enable them to integrate into the cell membrane. This structural feature is crucial for the receptor’s functionality, as it allows the receptor to perceive external signals and transmit them intracellularly. KD is typically a serine/threonine kinase responsible for catalyzing the transfer of phosphate from ATP to specific substrate proteins. This phosphorylation process is a critical post-translational modification that regulates various cellular functions and signaling pathways. Moreover, it has been demonstrated that LRR-RLKs with short extracellular domains primarily function as co-receptors. They assist in binding the ligand and work in conjunction with the ligand-binding receptor to stabilize and enhance the transduction of intracellular signals. Additionally, some LRR-RLKs can interact with one another, facilitating the formation of various heterodimers or trimers, which enables them to exhibit multifunctional capabilities. For example, BAK1 acts as a co-receptor, interacting with several LRR-RLKs, including FLS2, BRI1, and EFR, to facilitate downstream signaling [12].

Throughout the course of plant evolution, LRR-RLKs have diversified into a substantial gene family, reflecting their critical roles in various physiological processes. In *Arabidopsis thaliana*, approximately 200 LRR-RLK genes have been identified [13]. Similarly, rice (*Oryza sativa*) possesses around 300 LRR-RLK genes [13,14]. Wheat (*Triticum aestivum*) boasts a significantly larger family, with about 600 LRR-RLK genes [14]. Maize (*Zea mays*) also contains approximately 600 LRR-RLK genes [14]. Furthermore, the soybean (*Glycine max*) contains over 400 LRR-RLK genes [15,16] while the tomato (*Solanum lycopersicum*) has around 200 LRR-RLK genes [17]. The variation in the number of LRR-RLK genes across different species highlights the extensive evolutionary adaptation of these kinases, which are essential for mediating a range of physiological and developmental processes in plants.

Foxtail millet (*Setaria italica*), a drought- and salt-tolerant cereal crop domesticated in East Asia over 8500 years ago, has gained prominence as a pivotal model system for C_4_ plant biology and grass genomics. Its compact diploid genome, short life cycle, and exceptional abiotic stress resilience not only make it an ideal candidate for elucidating C_4_ photosynthetic evolution and environmental adaptation mechanisms but also establish it as a strategic genetic reservoir for improving climate-resilient crops. Phylogenetically nested within the Panicoideae subfamily, foxtail millet exhibits remarkable synteny with bioenergy staples like sugarcane, sorghum, maize, and rice, enabling cross-species translational research to enhance both food security and sustainable biofuel production [18]. This dual significance as an agronomic staple and a functional genomic bridge positions foxtail millet at the forefront of plant systems biology and agricultural innovation. In recent years, significant progress has been made in the genomic research of foxtail millet, including the successful construction of a haplotype map encompassing 916 core foxtail millet accessions [19] and a pan-genome variation map of the Setaria genus, which includes 113 representative foxtail millet resources [20]. However, our understanding of LRR-RLKs in foxtail millet remains limited.

## 2. Results

### 2.1. Classification of LRR-RLK Family Types in Foxtail Millet Genome

To investigate the distribution and function of the LRR-RLK family in foxtail millet, we conducted a search in the *Arabidopsis thaliana* database and identified 225 LRR-RLK family member protein sequences. These 225 Arabidopsis LRR-RLK protein sequences were used to perform a BLAST comparison against the millet genome database. We utilized the online tool at http://smart.embl-heidelberg.de/smart/show_motifs.pl, (accessed on 12 August 2023). for domain prediction, ensuring that only proteins containing at least one LRR domain and one protein kinase domain were classified as true LRR-RLKs. Redundant sequences lacking either the LRR domain or the protein kinase domain were removed. Ultimately, we identified a total of 285 protein sequences with the LRR-RLK structure in the millet genome, with their encoded amino acid lengths ranging from 462 to 1581 residues. Among these, 28 protein sequences lacked transmembrane regions, and 67 sequences did not possess signal peptides. Notably, the proteins encoded by the genes Seita.4G205200 and Seita.8G199200 were predicted to have two kinase domains.

Foxtail millet LRR-RLK sequences were classified into subfamilies based on their orthology to *Arabidopsis thaliana*. Using the predefined Arabidopsis LRR-RLK subfamily clades, we assigned millet sequences to corresponding subfamilies if they clustered within high-confidence clades in the phylogenetic tree (Figure 1). This analysis allowed us to categorize the LRR-RLK types within the foxtail millet genome. Among the identified sequences, Seita.7G281600, Seita.2G079600, Seita.3G406000, and Seita.9G458500 were categorized as ambiguous due to their unclear classification. The 285 LRR-RLK genes were classified into 14 types: LRR I (18), LRR II (15), LRR III (35), LRR IV (3), LRR V (9), LRR VI (9), LRR VII (8), LRR VIII-1 (7), LRR VIII-2 (11), LRR IX (3), LRR X (22), LRR XI (45), LRR XII (100).

A phylogenetic tree was constructed for the LRR-RLK family members in foxtail millet (Figure 2). The LRR-RLK family in foxtail millet can be divided into five branches. Notably, LRR XII is subdivided into two sections: one section, comprising 12 members, forms a branch with LRR XI, while the other section consists of 88 members. Additionally, LRR III and LRR VII share a branch, and LRRX is distinct in its own branch. The final branch is more complex, encompassing the remaining seven subfamilies. Among all subfamily members, LRR XII constitutes a significant proportion at 35.09%, followed by LRR XI and LRR III at 15.79% and 12.28%, respectively. The remaining types of each account for less than 10%, with the smallest subfamilies, LRR IV and LRR IX, each consisting of only three members. Through calculations of Ks, π, and dA values, it was estimated that this gene expansion occurred approximately 4.6 to 10.7 million years ago.

### 2.2. Distribution of the LRR-RLK Family in Foxtail Millet

The chromosomal distribution of LRR-RLK genes was determined by mapping their genomic coordinates to the nine chromosomes. Gene counts per chromosome were statistically analyzed and visualized (Figure 3). Among the 285 identified LRR-RLK family members in the foxtail millet genome, 283 genes are distributed across the nine chromosomes, except for Seita.J023100 and Seita.J030600. This might be due to the problem of chromosome assembly quality, resulting in the failure to determine the chromosomal locations of these two genes. The distribution of LRR-RLK family members on the chromosomes is not uniform (Table 1), with proportions ranging from 7.42% to 13.78%. Members of the LRR II, LRR III, LRR X, LRR XI, and LRR XII subfamilies are found across all nine chromosomes. Notably, the LRR XII subfamily has the largest number of members, with a significant concentration on chromosome 8, surpassing the distribution of other subfamilies (Figure 3).

We further analyzed gene clusters in each foxtail millet chromosome that contain more than five genes. Eight of these clusters are formed exclusively by members of the LRR I, LRR VIII-2, LRR X, LRR XI, and LRR XII subfamilies, while two additional clusters consist of LRR VIII-1/LRR III and LRR I/LRR VIII-2 (Figure 3). Notably, the four gene clusters of the LRR XII subfamily are distributed across chromosomes 1, 4, 5, and 8, indicating that this subfamily is relatively active within the foxtail millet genome.

Based on the physical locations of 283 LRR-RLK family members within the foxtail millet genome, we mapped them onto the nine chromosomes of foxtail millet (Figure 4). The results indicate that several foxtail millet LRR-RLK family members are clustered together on the chromosomes, forming gene clusters. Among the 283 genes in the millet LRR-RLK gene family, 139 genes are distributed within 44 gene clusters, accounting for 49.12% of the total. 

In cereal crops such as maize, rice, and sorghum, many LRR genes with similar functions are clustered together in close clusters. For instance, a gene cluster on chromosome 11 of rice contains 56 disease resistance genes [21]. The seventh chromosome of millet is homologous to chromosome 11 of rice, suggesting that the RLK genes located on the ninth chromosome of millet may possess disease resistance functions.

### 2.3. Transcriptome Analysis of the LRR-RLK Family in Foxtail Millet

An analysis of the transcriptome of the foxtail millet LRR-RLK family revealed significant differences in expression levels among the various family members (Figure 5). For example, members of the LRR III family exhibited higher expression levels in the millet ear compared to other tissues, while LRR II family members had relatively high expression levels in leaf and stem tissues. This indicates that different LRR-RLK subfamilies play distinct roles during the growth and development of millet. However, the widely distributed LRR XII subfamily in the millet genome showed generally low expression levels. This may be attributed to the presence of redundant genes generated during the expansion of this gene family or to their potential role as inducible expression genes, activated by external factors. For instance, among the 40 differentially expressed genes in the rice blast inoculation study, 22 belonged to the LRR XII subfamily. Certain subfamily members had exceptionally high expression levels. For instance, the LRR V family gene Seita.9G250400 had FPKM values of 79.86 in the stem and 76.42 in the root. Additionally, we identified homologous genes of the foxtail millet LRR-RLK family in rice and searched for differentially expressed genes in the RNA-Seq database [22] and the transcriptome chip database [23] following inoculation with rice blast disease (See Appendix A).

### 2.4. Functional Predictions for LRR-RLKs in Foxtail Millet Related to Stress Responses

Based on homology alignment with the LRR-RLKs functionally characterized in model plants, we predicted foxtail millet genes would potentially be involved in biotic and abiotic stress responses. For biotic stress (Table 2), foxtail millet LRR-RLKs are proposed to recognize pathogen-associated molecular patterns (PAMPs), such as bacterial flagellin, fungal chitin, or viral components, analogous to their homologs in Arabidopsis and rice, thereby activating defense responses including reactive oxygen species (ROS) production, defense gene induction, and co-receptor signaling cascades. For abiotic stress (Table 3), foxtail millet LRR-RLKs are hypothesized to mediate drought tolerance via ABA-dependent stomatal regulation, salt stress adaptation through ion homeostasis modulation, temperature stress responses via thermosensory signaling, and oxidative stress mitigation by enhancing ROS scavenging pathways.

## 3. Discussion

The comprehensive identification and classification of 285 LRR-RLK genes in foxtail millet, distributed across 14 subfamilies, underscores the expansion and diversification of this gene family in cereal genomes. In our study, the millet LRR XII subfamily contains 100 genes, with both rice and millet LRR XII members being significantly more numerous than those in Arabidopsis. This suggests that gene expansion in this subfamily may have occurred prior to the divergence of rice and millet. Conversely, the number of LRR I genes in Arabidopsis (45) far exceeds that in millet (18), indicating differing evolutionary trajectories between these species. Tandem duplications are a major mechanism of gene family expansion [24], leading to the formation of gene clusters on chromosomes. The predominance of the LRR XII subfamily (35.09% of all members) aligns with observations in other grasses, where lineage-specific duplication events, particularly tandem and segmental duplications, have driven the expansion of stress-related gene families. Notably, the colinearity between the foxtail millet chromosome 7 and rice chromosome 11, which harbors disease resistance gene clusters, implies conserved genomic organization for stress adaptation in grasses. The retention of clustered LRR-RLKs, such as those on millet chromosome 9, may reflect selective pressure to maintain pathogen recognition capabilities, consistent with the enrichment of LRR XII members among differentially expressed genes during rice blast infection. The structural and phylogenetic diversity of LRR-RLKs in foxtail millet suggests their multifunctional roles in mediating critical responses to both biotic challenges—such as pathogen invasion—and abiotic adversities, including drought, salt, temperature extremes, and oxidative stress.

### 3.1. LRR-RLKs’ Function in Plant Immunity

Plants rely heavily on innate immunity to defend themselves against a wide range of phytopathogens. In foxtail millet production practices, various diseases adversely affect both yield and quality. Common and impactful diseases include foxtail millet rust, downy mildew, foxtail millet blast, and foxtail millet sheath blight, among others. Immune responses in plants are triggered upon recognition of microbial molecular patterns by pattern recognition receptors (PRRs) [25,26]. It is now evident that RLKs serve as crucial PRRs for microbe- and plant-derived molecular patterns indicative of pathogen incursion. To date, several RLK–ligand pairs have been established. Rice RLK protein Xa21, which confers resistance against diverse Xanthomonas spp., was the first RLK whose encoding gene was cloned [27]. The Arabidopsis RLK protein FLS2 recognizes a conserved 22-amino acid (flg22) from the N terminus of the bacterial flagellin, and activates the plant’s basic immune response, including the generation of reactive oxygen species and the induction of defense gene expression [28]. Another Arabidopsis RLK, EFR, senses a conserved N-terminal peptide sequence of the bacterial elongation factor-Tu, termed elf18 [6,29]. The Arabidopsis CERK1, an RLK with three Lysine motifs (LysMs) in its ectodomain, is necessary for the recognition of chitin, a main component of the fungal cell wall, via direct binding [30,31,32,33]. In rice, CEBiP, an RLP protein containing two LysMs, is the high-affinity chitin receptor [34,35,36,37]. Arabidopsis LYM1 and LYM3 physically bind Peptidoglycans (PGNs), a major component of cell walls of both Gram-positive and Gram-negative bacteria, to mediate resistance to bacterial pathogens [38]. It has also been reported that LysM RLPs play a role in PGN perception in rice. LYP4 and LYP6 in rice can bind both PGNs and chitin, thus functioning as dual-function receptors to defend against both bacteria and fungi [39]. The Arabidopsis RLK protein NUCLEAR SHUTTLE PROTEIN-INTERACTING KINASE1 (NIK1) is a key pattern recognition receptor that mediates plant antiviral immunity by detecting viral PAMPs, triggering immune responses, and activating transcription factors [40]. Moreover, BRI1-ASSOCIATED KINASE 1 (BAK1) [12], CHITIN ELICITOR RECEPTOR KINASE 1 (CERK1) [30], and SUPPRESSOR OF BIR1-1 (SOBIR1) [41] are common co-receptors that play a significant role in microbial molecular pattern-triggered immunity. These co-receptors are evolutionarily conserved in various plant species [42,43]. Through comparative analysis with reported disease resistance genes in other crops, it is possible to predict disease-resistance-related genes in foxtail millet.

### 3.2. LRR-RLKs’ Function in Drought Stress

Drought is a prevalent environmental stress and has harmful effects on plant growth and development. Although foxtail millet is primarily cultivated in arid and semi-arid regions and exhibits inherent drought tolerance, extreme drought conditions still severely impact its agricultural productivity. Therefore, conducting a comparative analysis of drought-resistant *LRR-RLK* genes in other crops can effectively predict drought-tolerant LRR-RLK genes in foxtail millet. ABA serves as a crucial regulator in plant responses to drought stress, affecting the expression of osmotic-stress-responsive genes, facilitating adaptive physiological changes, and shaping plant growth [44]. A recent study found that the CLAVATA3/EMBRYO-SURROUNDING REGION-RELATED 25 (CLE25) peptide is transported from root vascular tissues to leaves in response to decreasing soil water availability, where it is recognized by the RLK BARELY ANY MERISTEM (BAM), triggering a drought signal that induces the expression of the ABA biosynthetic enzyme NINE-CIS-EPOXYCAROTENOID DIOXYG-ENASE 3 (NCED3) [45]. The *ERECTA* (*ER*) gene (from the Latin *erectus*, meaning “upright”) is a canonical leucine-rich repeat receptor-like kinase (LRR-RLK) first identified in *Arabidopsis thaliana* for its role in regulating organ elongation, stomatal patterning, and inflorescence architecture. Mutations in *ER* result in compact inflorescences and shortened internodes. The LRR-RLK ER negatively regulates stomatal density and plays a crucial role in controlling transpiration efficiency [46]. Notably, the overexpression of ER has been shown to enhance drought tolerance in various plants without incurring a yield penalty [47].

RLKs have been proven to play a key role in the response to drought stress mediated by ABA. FLORAL ORGAN NUMBER1 (FON1), an RLK protein, has been reported to regulate drought stress and early growth processes by modulating ABA signaling in rice, specifically by mediating the expression of stress-related and ABA-responsive genes associated with the accumulation of reactive oxygen species (ROS) [48]. An RLK gene named Leaf Panicle 2 (LP2) serves as a negative regulator in the drought response. The overexpression of LP2 in rice results in reduced accumulation of H₂O₂, increased stomatal openness in leaves, and heightened sensitivity to drought stress [49]. The Arabidopsis RLK gene LRK10L1.2 acts as a positive regulator of drought tolerance by facilitating stomatal closure, potentially through direct or indirect mechanisms involving ABA-mediated signaling [50]. Stomata play a crucial role in plant drought resistance by closing during drought conditions to reduce water loss. The rice RLK proteins OsSIK1 [51] and OsSIK2 [52] suppress stomatal development and reduce stomatal density in rice leaves, thereby enhancing drought tolerance by minimizing water loss. A recent study demonstrated that KIN7 and its close homolog, LRR1 (LEUCINE RICH REPEAT PROTEIN 1), are targeted for degradation by PLANT U-BOX 11 (PUB11) under conditions of abscisic acid (ABA) or drought stress [53]. This finding underscores the important regulatory role that RLKs play in the plant’s response to drought stress.

### 3.3. LRR-RLKs’ Function in Salt Stress

Excessive salt, which severely impacts foxtail millet yield, is a seriously limiting factor in plant growth and development. The entire processes of perception, signaling, and response to the salinity stress can be outlined as the osmosensory and ion-sensory proteins in the cell wall and the plasma membrane, respectively, perceiving salinity in the form of osmotic and ionic stress [54]. Many RLKs are induced by reactive oxygen species (ROS) levels and aim to control the ROS-mediated damage during salinity stress. For instance, for the plasma membrane RLK protein, *Pisum sativum* lectin receptor-like kinases (PsLecRLK), its transcripts were upregulated when under salinity stress. The *PsLecRLK*-overexpressed plants exhibited greater tolerance to salt stress due to the ROS-scavenging enzymes, which led to reduced ROS accumulation and, consequently, less membrane damage [55]. The rice RLKs, OsSlk1 and OsSTLK, are induced by salt stress and reduce the ROS load and lipid oxidation [56]. In *Arabidopsis thaliana*, SIT1 promotes the accumulation of ROS, resulting in growth inhibition and plant death under salt stress, and this occurs in a manner dependent on MPK3/6 and ethylene signaling [57]. Some RLKs spanning the cell wall recognize peptide hormones expressed due to salinity stress. For instance, PepR1, an RLK protein in Arabidopsis, plays a vital role in salinity stress tolerance by recognizing peptide hormones like AtPep3, which are expressed in response to salt stress [58]. Salt-stress-induced damage to the cell wall and cell–cell interactions is sensed by an RLK named FERONIA (FER), which belongs to the *Catharanthus roseus* RLK subfamily (CrRLK). FER has also been shown to alter calcium signaling and regulate specific transcriptional responses to mechanical perturbation [59,60]. The FER loss-of-function mutant exhibited hypersensitivity to both ABA and salt stress, suggesting that FER may regulate the salt stress response via the ABI2-mediated ABA signaling pathway [61]. In addition, OsRMC encodes an RLK that functions as a negative regulator of salt stress responses in rice. Its transcription is significantly induced by salt treatment and exhibits a dose-dependent response to stress [62]. The comparative analysis of salt-tolerant genes between foxtail millet and other crops enables the accurate prediction of salinity-resistant genes in this cereal, thereby informing molecular breeding strategies for enhancing salt tolerance.

### 3.4. LRR-RLKs’ Function in Temperature Stress

Temperature is a critical environmental factor that significantly impacts various aspects of plant metabolism, growth, and development. Climate change, along with the resulting extreme temperature fluctuations, poses a major threat to plant growth and resilience [63]. The fluidity of cellular membranes can be either reduced or enhanced in response to cold or heat stress. These changes can be detected by membrane-localized RLKs [64]. Calcium-mediated signaling is thought to play a crucial role in plant responses to both cold and heat stress. Several calcium-regulated genes have been identified and validated in this context. Arabidopsis CALCIUM/CALMODULIN-REGULATED RECEPTOR-LIKE KINASES 1 (CRLK1) and 2 (CRLK2), which are calcium-regulated receptor-like kinases (RLKs), play a crucial role in cold tolerance in plants. They positively regulate cold stress responses by suppressing the cold-induced activity of MPK3 and MPK6 [65]. The RLK protein COLD-RESPONSIVE PROTEIN KINASE 1 (CRPK1) regulates key cold-responsive C-repeat-binding factor (CBF) proteins by phosphorylating 14-3-3 proteins, which then shuttle from the cytosol to the nucleus to interact with and destabilize the CBF proteins [66]. In rice, *COLD TOLERANCE AT BOOTING STAGE 4a* (*CTB4a*) encodes a conserved RLK that interacts with AtpB, a beta subunit of ATP synthase. This interaction enhances ATP synthase activity, leading to improved seed setting and increased yield under cold stress conditions [67]. The Arabidopsis RLK protein PHLOEM INTERCALATED WITH XYLEM-LIKE 1 (AtPXL1) was found to be highly induced by both cold and heat stress. AtPXL1 exhibits autophosphorylation activity, and the atpxl1 mutant displayed hypersensitive phenotypes during germination under cold and heat stress conditions. These findings suggest that AtPXL1 plays a positive role in mediating signal transduction in response to cold and heat stress [68]. GsLRPK, an RLK gene from *G. soja*, was specifically and quickly activated in response to cold [69]. However, its functions require further study. Additionally, as we mentioned above, the overexpression of ERECTA not only improves drought tolerance but also enhances thermotolerance in Arabidopsis, rice, and tomato plants [46,47,70]. In rice, the RLKs 25L1 and 25L2 are involved in mediating hybrid weakness under high temperature conditions, while the two RLKs, THERMO-SENSITIVE GENIC MALE STERILE 10 (TMS10) and TMS10-LIKE (TMS10L), are responsible for regulating male fertility in rice at elevated temperatures [71,72]. The comparative analysis of temperature-responsive genes between foxtail millet and other crops provides predictive insights into thermal-adaptation-related genes in this cereal, thereby facilitating molecular breeding for enhanced thermotolerance and cold resistance.

### 3.5. LRR-RLKs’ Function in Oxidation Stress

Oxidative stress occurs when the production of reactive oxygen species (ROS) surpasses the capacity of the cellular antioxidant defense system. ROS are a diverse group of highly reactive molecules, including singlet oxygen (O_2_), superoxide anions (O^2−^), hydrogen peroxide (H_2_O_2_), and hydroxyl radicals (HO·). Increasing evidence suggests that RLKs play a role in mediating the oxidative stress in plants. In Arabidopsis, apoplastic H_2_O_2_ can be directly sensed by an RLK named HYDROGEN PEROXIDE-INDUCED Ca^2+^ INCREASES1 (HPCA1). H_2_O_2_-activated HPCA1 induces increases in Ca^2+^ in guard cells and promotes stomatal closure [73]. GUARD CELL HYDROGEN PEROXIDE-RESISTANT1 (GHR1) encodes an RLK localized on the plasma membrane in *Arabidopsis thaliana*. The ghr1 mutant exhibits defects in the H_2_O_2_-regulated activation of S-type anion currents in guard cells, indicating that GHR1 is involved in the regulation of stomatal movement in response to oxidative stress [74]. Another example involves the extracellular GRIM REAPER (GRI) protein. Upon the accumulation of ROS in the apoplast, GRI is cleaved by METACASPASE9. The resulting 11-amino acid GRI peptide binds to PRK5 to initiate cell death [75]. The comparative analysis of oxidative-stress-responsive genes between foxtail millet and other crops enables the systematic prediction of key regulatory genes, thereby facilitating molecular breeding for improved oxidative stress tolerance in this cereal.

The analysis of the role played by the LRR-RLK (leucine-rich repeat receptor-like kinases) gene family in plant environmental response mechanisms reveals its diversity and complexity in regulating physiological processes and responding to environmental stresses. As a stress-tolerant and barren-tolerant crop, foxtail millet serves as a valuable model for studying the functions of the LRR-RLK gene family under abiotic stress conditions. The comparative genomic analysis of foxtail millet identified potential stress-responsive LRR-RLK genes through homology-based screening. Using BLASTp (https://phytozome-next.jgi.doe.gov/blast-search, accessed on 17 August 2023) and domain validation, we retrieved homologs of experimentally verified stress-related genes from model plants. We identified the genes in foxtail millet that may be involved in biotic stress (Table 2) and abiotic stress (Table 3). This can provide some clues for the stress resistance research of foxtail millet. However, further experimental validation is required to elucidate the precise biological mechanisms.

This study provides a comprehensive identification and functional prediction of the LRR-RLK gene family in foxtail millet, laying a solid foundation for understanding how plants respond to biotic and abiotic stresses. It also offers significant insights for future crop improvement and stress resistance research. Nonetheless, numerous unanswered questions and potential research areas remain to be further investigated.

### 3.6. Perspectives

The study of the LRR-RLK gene family represents a critical breakthrough in understanding plant stress response mechanisms and offers valuable resources for breeding stress-resistant crops. Future research should prioritize multidimensional exploration: First, conduct gene expression analyses across diverse plant species or within the same species. Based on the spatiotemporal expression characteristics of genes, researchers should design targeted stress-resistant breeding strategies. For instance, this study analyzed the expression patterns of foxtail millet *RLK-LRR* genes in the spical, leaves, stems, and roots, and performed homologous comparison analyses with rice genes (Appendix A). This approach provides a template for leveraging evolutionary conservation in crop improvement programs. Second, functional validation through CRISPR/Cas9 editing, RNA interference, and overexpression analysis is essential to confirm the roles of these genes in stress responses, particularly addressing redundancy and inducible expression patterns in the significantly expanded LRR XII subfamily. Third, dissecting their regulatory networks requires integrating multi-omics approaches to map interactions with MAPK pathways and hormone signals, thereby elucidating systemic signal transduction mechanisms. Fourth, comparative genomic analyses across Poaceae crops like foxtail millet, rice, and wheat will reveal subfamily-specific expansions as adaptive strategies to environmental pressures, while systematic comparisons with Arabidopsis could uncover conserved regulatory elements. Fifth, agricultural applications hold great promise for transferring drought-/salt-tolerant LRR-RLK genes into staple crops via gene editing, developing molecular markers for marker-assisted selection through transcriptomic/metabolomic integration, and engineering metabolic pathways for enhanced stress resistance. Lastly, investigating multi-stress response mechanisms through simulated environmental models will clarify functional differentiation among family members under complex stress conditions. By synergizing multidisciplinary approaches (genomics, systems biology) with cutting-edge technologies (intelligent breeding, gene editing), this research framework aims to unlock the full potential of LRR-RLK genes for ensuring global food security and adapting to climate change challenges.

## 4. Materials and Methods

### 4.1. Acquisition of LRR-RLK Family Sequences in Foxtail Millet

We initiated the study by the sequence names of the LRR-RLK protein subfamilies from the *Arabidopsis thaliana* database. Using gene IDs as search keywords, we retrieved the LRR-RLK protein sequences from the TAIR v10.0 database (http://www.arabidopsis.org/, accessed on 12 August 2023) and classified them according to GenBank entries for Arabidopsis LRR-RLK identification (accession numbers: FJ708625-FJ708818.) [76]. To identify the corresponding sequences in foxtail millet, we obtained the LRR-RLK protein sequences from the millet genome using the Phytozome V12.1 database (https://phytozome.jgi.doe.gov/pz/portal.html, accessed on 12 August 2023)). Phytozome is a publicly accessible plant genomics database maintained by the Joint Genome Institute (JGI), which hosts annotated genomes of various species, including foxtail millet. We employed BLAST analysis with the Arabidopsis LRR-RLK protein sequences to identify the corresponding sequences in millet (E-value cutoff of < 1 × 10^−10^). After organizing the downloaded protein sequences, we performed domain predictions using the online tool at (http://smart.embl-heidelberg.de/smart/show_motifs.pl, accessed on 17 August 2023); only proteins containing at least one LRR domain and one protein kinase domain were classified as true LRR-RLKs, and redundant sequences lacking these domains were removed.

### 4.2. The Classification of LRR-RLK Family Sequences in Foxtail Millet

We constructed a phylogenetic tree for both the Arabidopsis and foxtail millet LRR-RLK protein sequences using MEGA 6.0 software [77]. Initially, we generated a phylogenetic tree that included sequences from both species, utilizing ClustalX for sequence alignment with default parameters, and employed the Maximum Likelihood method for tree construction. This approach allowed us to classify the foxtail millet LRR-RLK family based on the well-established classification of the Arabidopsis LRR-RLK family. The evolutionary tree of foxtail millet LRR-RLK family members was constructed using the Maximum Likelihood method, with a bootstrap value set to 500 and other parameters maintained at default values.

### 4.3. Analysis of LRR-RLK Family Structure in Foxtail Millet

We identified for the physical location of the LRR-RLK family genes in foxtail millet using the Phytozome12.1 database. By comparing the gene splicing patterns of the LRR-RLK family members predicted in Phytozome12.1, the number and position of introns and exons in each LRR-RLK member were determined. The chromosomal distribution map of LRR-RLK genes was generated using the online tool MG2C (MapGene2 Chromosome v2.1) (http://111.203.21.71:8000/multi-omics/transcriptome.html, accessed on 17 August 2023). For visualization, a single chromosome container was configured with a width of 350 pixels, while other parameters were retained as default settings [78,79]. The software automatically plotted the physical positions of LRR-RLK genes across the nine foxtail millet chromosomes.

### 4.4. Transcriptome Analysis of LRR-RLK Family in Foxtail Millet

We download transcriptome data of LRR-RLK family members identified in the foxtail millet genome (http://111.203.21.71:8000/multi-omics/transcriptome.html, accessed on 17 March 2025). Plants were grown under a 16/8 h light/dark photoperiod, 28 °C/22 °C day/night temperatures, and 50–60% relative humidity to mimic natural seasonal variations. We analyzed the differential transcriptional expression levels of these members. Additionally, we identified homologous genes of the foxtail millet LRR-RLK family in rice and searched for differentially expressed genes in the RNA-Seq database [22] and transcriptome chip database [23] following inoculation with rice blast disease. Grouped cluster heatmaps were generated using the CNSKnowall online software (https://cnsknowall.com, accessed on 17 March 2025) to visualize expression patterns.

## Figures and Tables

**Figure 1 ijms-26-04576-f001:**
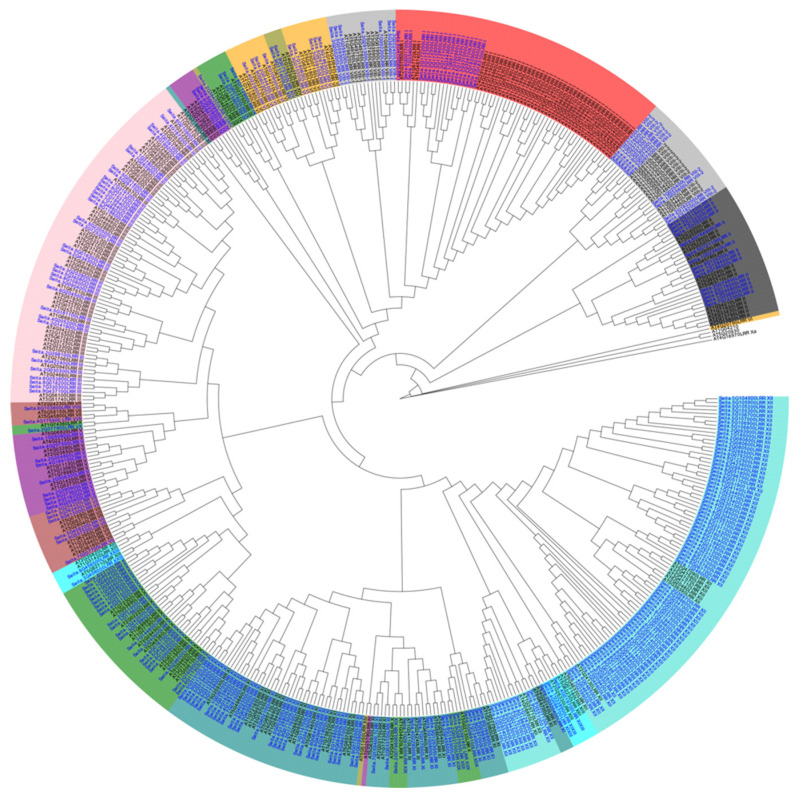
The evolution tree of LRR-RLK members in Arabidopsis and foxtail millet. In the phylogenetic tree, the colors are defined as follows: for the font color, black denotes *Arabidopsis thaliana* and blue denotes *Setaria italica*; for the background color, different shades represent LRR classification types (e.g., dark blue = LRR XI; light blue = LRR XII). Branch length represents evolutionary distance.

**Figure 2 ijms-26-04576-f002:**
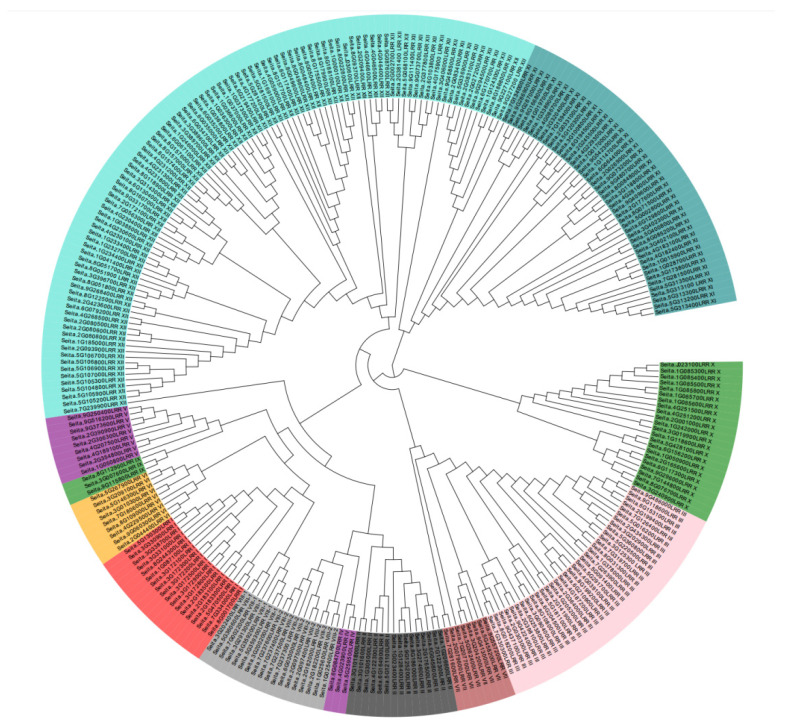
The evolution tree of LRR-RLK members in foxtail millet. Color gradients denote subfamily classification (e.g., dark blue = LRR XI; light blue = LRR XII). Branch length represents evolutionary distance.

**Figure 3 ijms-26-04576-f003:**
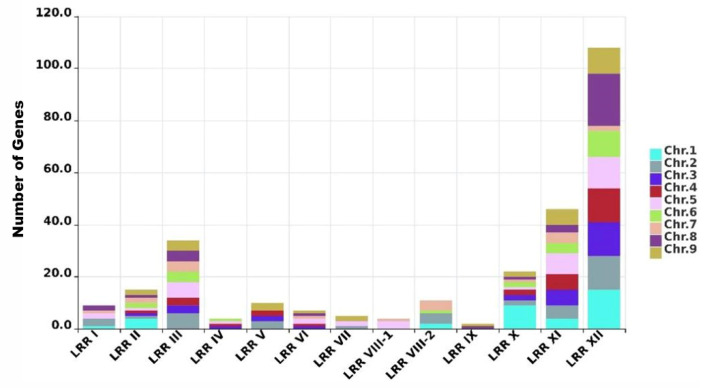
The gene distribution of the LRR-RLK subfamily on chromosomes.

**Figure 4 ijms-26-04576-f004:**
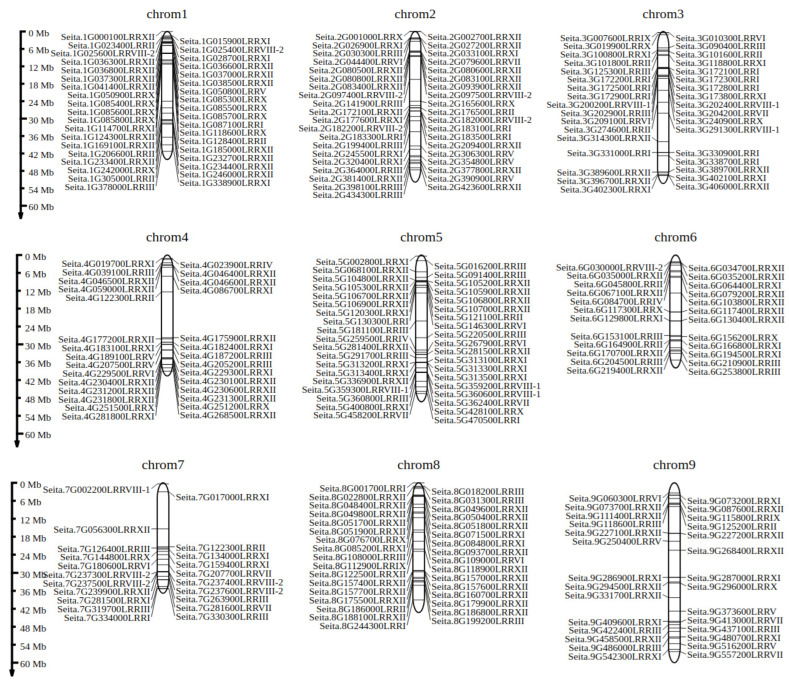
The distribution of LRR-RLK genes represented in 9 chromosomes of foxtail millet.

**Figure 5 ijms-26-04576-f005:**
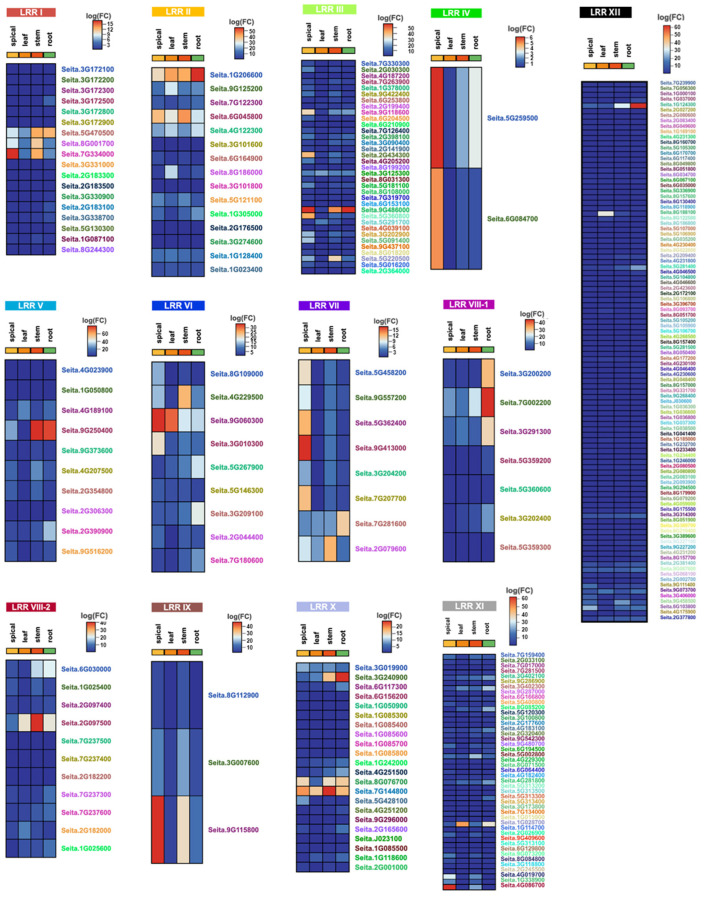
The FPKM of LRR-RLK family members in foxtail millet.

**Table 1 ijms-26-04576-t001:** The distribution of LRR-RLK gene clusters on chromosomes.

Chr.	I	II	III	IV	V	VI	VII	VIII	IX
Number of LRR-RLK genes	37	39	34	29	38	24	21	33	28
Number of LRR-RLK genes in the gene cluster	19	21	21	17	22	9	9	13	8
Concentration of LRR-RLK genes in the gene cluster (%)	51.35	53.85	61.76	58.62	57.89	37.50	42.86	39.39	28.57
Concentration of LRR-RLK genes on the chromosome (%)	13.07	13.78	12.01	10.25	13.43	8.48	7.42	11.66	9.89

**Table 2 ijms-26-04576-t002:** LRR-RLKs involved in biotic stress.

Gene ID.	Gene Name	Plant Origin	Function	Homologous Genes in *Setaria italica*
Os04g0226800	*Xa21*	*Oryza sativa*	Bacterial recognition and immune response	Seita.7G056300/Seita.7G056900
At5g46330	*FLS2*	*Arabidopsis thaliana*	PAMP recognition, initiation of immune response (recognizes bacterial flagellin)	Seita.1G161700 Seita.1G169100/Seita.6G071500/Seita.7G239900
At5g20480	*EFR*	*Arabidopsis thaliana*	PAMP recognition, immune response (recognizes bacterial EF-Tu)	Seita.1G227300 Seita.1G233300/Seita.1G233400/Seita.1G234400/Seita.2G016400/Seita.3G354900/Seita.3G389600/Seita.5G106400/Seita.5G107200/Seita.6G117500/Seita.8G051700/Seita.8G157300/Seita.8G157400
At3g21630	*CERK1*	*Arabidopsis thaliana*	Chitin recognition and fungal immunity response	Seita.2G269700/Seita.5G280500/Seita.6G227400
Os03g0133400	*CEBIP(RLP)*	*Oryza sativa*	Chitin oligosaccharide recognition	Seita.9G550700
AT1G21880	*LYM1*	*Arabidopsis thaliana*	Peptidoglycan perception	Seita.1G333700/Seita.2G226100/Seita.4G084200/Seita.9G011000
AT1G77630	*LYM3*	*Arabidopsis thaliana*	Peptidoglycan perception	Seita.1G333700
Os09g27890	*LYP4*	*Oryza sativa*	Dual recognition of chitin and peptidoglycan	Seita.2G226100/Seita.5G280500
Os06g10660	*LYP6*	*Oryza sativa*	Dual recognition of chitin and peptidoglycan	Seita.4G084200
AT5G16000	*NIK1*	*Arabidopsis thaliana*	Antiviral defense signaling	Seita.1G305000/Seita.4G122300/Seita.4G122500
At4g33430	*BAK1*	*Arabidopsis thaliana*	BR signaling, PAMP-triggered immunity, regulation of cell death	Seita.6G045800/Seita.1G206600/Seita.7G122300/Seita.4G122300
At2g31880	*SOBIR1*	*Arabidopsis thaliana*	Involved in immune response, forms complexes with other receptors to perceive PAMPs	Seita.1G207200 Seita.2G076800/Seita.4G126700/Seita.4G126800
EU041719	*LecRLK*	*Pisum sativum*	Pathogen recognition	Seita.9G240100
AT5G16590	*LRR1*	*Arabidopsis thaliana*	Defense response regulation	Seita.7G126400/Seita.9G118600/Seita.3G202900/Seita.3G202900
At1g73080	*PEPR1*	*Arabidopsis thaliana*	DAMP recognition, immune response (recognizes plant endogenous peptides)	Seita.6G166800 Seita.6G167000/Seita.9G287000

**Table 3 ijms-26-04576-t003:** LRR-RLKs involved in abiotic stress.

Gene ID.	Gene Name	Plant Origin	Function	Homologous Genes in *Setaria italica*
AT2G26330	*ERECTA*	*Arabidopsis thaliana*	Organ development and patterning regulation	Seita.1G084800/Seita.1G338900/Seita.4G086700/Seita.5G281200
AT3G23130	*FON1*	*Arabidopsis thaliana*	Floral meristem regulation	Seita.1G000900
AT2G38530	*LP2*	*Arabidopsis thaliana*	Development and defense response	Seita.7G301200/Seita.7G301100/Seita.8G013600/Seita.5G363000/Seita.8G013500/Seita.7G300900/Seita.3G204700
AT1G18390	*LRK10L1.2*	*Arabidopsis thaliana*	Stress response signaling	Seita.3G157100/Seita.3G157200/Seita.5G093200/Seita.5G093300/Seita.5G093600/Seita.5G093700/Seita.5G093900/Seita.5G094200/Seita.5G095000/Seita.5G095300/Seita.5G277200/Seita.5G277400
Os05g0461600	*SIK1*	*Oryza sativa*	Abiotic stress response	Seita.3G215600
Os07g0186200	*SIK2*	*Oryza sativa*	Abiotic stress response	Seita.3G040300/Seita.3G040400/Seita.3G040500/Seita.3G060400/Seita.3G060600
AT3G02880	*KIN7*	*Arabidopsis thaliana*	Kinase signaling pathway	Seita.9G118600/Seita.7G126400/Seita.7G126400/Seita.5G360800
Os06g0130100	*SLK1*	*Oryza sativa*	Salt stress response	Seita.4G019700
Os05g0305900	*STLK*	*Oryza sativa*	Salt tolerance signaling	Seita.3G291300
Os02g0640500	*SIT1*	*Oryza sativa*	Salt stress tolerance	Seita.1G249800
AT3G51550	*FER*	*Arabidopsis thaliana*	Growth and stress response regulation	Seita.3G288700/Seita.3G289000/Seita.7G215600/Seita.8G153000/Seita.9G381900/Seita.9G382300/Seita.9G382400/Seita.9G382500/Seita.9G382700/Seita.9G414400
BAF16049.1	*RMC*	*Oryza sativa*	Root growth regulation	Seita.9G250400/Seita.9G516200/Seita.2G390900

## Data Availability

Data is contained within the article and Appendix A.

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
