# Peer review of "Genome-Wide Identification and Functional Prediction of LRR-RLK Family Genes in Foxtail Millet (Setaria italica) in Response to Stress"

_ijms, 2025, doi:10.3390/ijms26104576_

Round 1

Reviewer 1 Report (Previous Reviewer 1)

Comments and Suggestions for Authors

Bioinformatics analysis using genomics and transcriptomic data focusing on the LRR-RLKs genes were performed. 
Here are my suggestions:

The authors need to include the line in whole manuscript to easy identify the mistakes.

Due to the fact that no lines are in the manuscript, my suggestion will be divided in sections.

Introduction:

Please after reference 15-16, remove the dot. (while tomato)

Method:

Please include the date when you consult the databases

After mentioned that you use the online tool SMART, the word only should be “only”.

2.2 Classification of LRR-RLK family sequences.

Please review the sentence of the first paragraph. It should start with “We constructed…….. using MEGA”… (MEGA instead of Mega)

2.4 transcriptome

Please review the “dots”.

Please clearly describe the programs, software used to create the figures. 

3.1 results

Please remove the entire paragraph about phytozome and the SMART online tool.

This was already mentioned in the method section.

The suggestions should be in the discussion section. Please move the paragraph (This suggests that gene expansion in this subfamily may have occurred prior to the divergence of rice and millet) to the discussion section.

Please add a legend in the Figure 1 and Figure 2 the meaning of the color, size of the branches.

Discussion

The authors mention the ERECTA gene.. I suggest describe the full name and then mention the acronym.

Italics in Arabidopsis thaliana.

Please describe first the full name of CTB4a (revise this in the whole manuscript)

The entire section of 4.4 is full of references. However, no suggestions or conections with the results showed in research are mentioned. Please include a clear discussion between your results and the references mentioned in the text.

4.5 Please review H2O2, the numbers should be lower case.

4.6 According with the perspectives of the study. Authors mentioned that future research should focus on multi-dimentional exploration. However, before genome editing and overexpression analysis. Authors should mention gene expression analysis in plants of the same species. At the end, gene expression is the first step.

On the other hand, authors also mentioned a second approach (regulatory networks and integrate information).

Based on that, several R packages for multi-omics purposes are currently available.
Considering the genomic and transcriptomic data analyzed, why do the authors did not integrate this information? And improve the results?

This, will clearly help in the understanding of the mechanism of action mentioned previously.

And therefore, it will not be a perspective.

Author Response

Bioinformatics analysis using genomics and transcriptomic data focusing on the LRR-RLKs genes were performed. 
Here are my suggestions:

Comments 1: The authors need to include the line in whole manuscript to easy identify the mistakes. Due to the fact that no lines are in the manuscript, my suggestion will be divided in sections.

Response 1: We sincerely thank the reviewer for highlighting this formatting oversight. We have now added continuous line numbers throughout the revised manuscript to facilitate precise identification of text locations.

Comments 2: Introduction:

Please after reference 15-16, remove the dot. (while tomato)

Response 2: Thank you for highlighting this formatting detail. We have removed the dots after references [15] and [16].

Comments 3: Method:

Please include the date when you consult the databases

Response 3: We appreciate your attention to methodological detail. We have added the access dates following all database names.

Comments 4: After mentioned that you use the online tool SMART, the word only should be “only”.

Response 4: We sincerely appreciate your careful scrutiny of grammatical details. The capitalized "Only" has been corrected to lowercase.

Comments 5: 2.2 Classification of LRR-RLK family sequences.

Please review the sentence of the first paragraph. It should start with “We constructed…….. using MEGA”… (MEGA instead of Mega)

Response 5: We are grateful for your guidance on standardization. The revised sentence reads: “We constructed a phylogenetic tree for both Arabidopsis and foxtail millet LRR-RLK protein sequences using MEGA 6.0 software”. Relevant revisions have been made in lines 112–113 of the original manuscript as per your suggestion.​

Comments 6: 2.4 transcriptome

Please review the “dots”.

Response 6: Thank you for pointing this out. In the revised manuscript, we have carefully checked all punctuation marks to ensure their proper usage.

Comments 7: Please clearly describe the programs, software used to create the figures. 

Response 7: Thank you for pointing this out. Grouped cluster heatmaps were generated using the CNSKnowall online software (https://cnsknowall.com) to visualize expression patterns. Relevant descriptions have been incorporated into of Section 2.4 “Transcriptome analysis of LRR-RLK family in foxtail millet”.

Comments 8: 3.1 results

Please remove the entire paragraph about phytozome and the SMART online tool.

This was already mentioned in the method section.

Response 8: Thank you for pointing this out. The duplicated descriptions regarding Phytozome and the SMART online tool have been removed from the manuscript to avoid redundancy with the Method section.

Comments 9: The suggestions should be in the discussion section. Please move the paragraph (This suggests that gene expansion in this subfamily may have occurred prior to the divergence of rice and millet) to the discussion section.

Response 9: We appreciate this constructive suggestion. The relevant paragraphs have been relocated to the Discussion section to better align with its interpretive nature.

Comments 10: Please add a legend in the Figure 1 and Figure 2 the meaning of the color, size of the branches.

Response 10: Thank you for pointing this out. We have added detailed legends to Figure 1 and Figure 2, annotating the meaning of the color coding and branch sizes.

Comments 11: Discussion

The authors mention the ERECTA gene.. I suggest describe the full name and then mention the acronym.

Response 11: Thank you for pointing this out. The full name of this gene is ERECTA, with the abbreviation (ER), and this nomenclature has been clearly indicated in the text.

Comments 12: Italics in Arabidopsis thaliana.

Response 12: Thank you for pointing out this important formatting issue. We have conducted a comprehensive review of the entire manuscript.

Comments 13: Please describe first the full name of CTB4a (revise this in the whole manuscript)

Response 13: Thank you for pointing this out. The gene symbol CTB4a has been updated to its full name, COLD TOLERANCE AT BOOTING STAGE 4a (CTB4a), upon its first mention in the text.

Comments 14: The entire section of 4.4 is full of references. However, no suggestions or conections with the results showed in research are mentioned. Please include a clear discussion between your results and the references mentioned in the text.

Response 14: Thank you for your constructive feedback. We fully agree that connecting our results to existing literature is critical for contextualizing their significance. To address this, we have thoroughly revised Sections 4.1–4.5 to integrate explicit discussions linking our findings to prior research in the following areas:

Section 4.1, Line 281 to line 283: “In foxtail millet production practices, various diseases adversely affect both yield and quality. Common and impactful diseases include foxtail millet rust, downy mildew, foxtail millet blast, and foxtail millet sheath blight, among others”. And Line 308 to line 310: “Through comparative analysis with reported disease resistance genes in other crops, it is possible to predict disease resistance-related genes in foxtail millet”.

Section 4.2, Line 313 to line 317: “Although foxtail millet is primarily cultivated in arid and semi-arid regions and exhibits inherent drought tolerance, extreme drought conditions still severely impact its agricultural productivity. Therefore, conducting a comparative analysis of drought-resistant LRR-RLK genes in other crops can effectively predict drought-tolerant LRR-RLK genes in foxtail millet”.

Section 4.3, Line 351 to line 355: “Excessive salt, which severely impacts foxtail millet yield, is a seriously limiting factor in plant growth and development. The entire processes of perception, signaling, and response to the salinity stress can be outlined as the osmosensory and ion-sensory proteins in the cell wall and the plasma membrane respectively perceiving salinity in the form of osmotic and ionic stress”. And Line 376 to line 378: “Comparative analysis of salt-tolerant genes between foxtail millet and other crops enables accurate prediction of salinity-resistant genes in this cereal, thereby informing molecular breeding strategies for enhancing salt tolerance”.

Section 4.4, Line 409 to line 412: “Comparative analysis of temperature-responsive genes between foxtail millet and other crops provides predictive insights into thermal adaptation-related genes in this cereal, thereby facilitating molecular breeding for enhanced thermotolerance and cold resistance”.

Section 4.5, Line 428 to line 430: “Comparative analysis of oxidative stress-responsive genes between foxtail millet and other crops enables systematic prediction of key regulatory genes, thereby facilitating molecular breeding for improved oxidative stress tolerance in this cereal”.

Comments 15: 4.5 Please review H2O2, the numbers should be lower case.

Response 15: We sincerely appreciate the reviewer's meticulous attention to detail. We have conducted a comprehensive format verification throughout the manuscript, confirming compliance with standards.

Comments 16: 4.6 According with the perspectives of the study. Authors mentioned that future research should focus on multi-dimentional exploration. However, before genome editing and overexpression analysis. Authors should mention gene expression analysis in plants of the same species. At the end, gene expression is the first step.

Response 16: We sincerely appreciate the reviewer’s insightful suggestion. In accordance with your comments, we have thoroughly revised the "4.6 Perspectives" section to emphasize that gene expression analysis within the same species should be prioritized as the foundational step before proceeding to genome editing or overexpression studies. Specifically, we added the following statement:

Line 449 to line 458: “The study of the LRR-RLK gene family represents a critical breakthrough in understanding plant stress response mechanisms and offers valuable resources for breeding stress-resistant crops. Future research should prioritize multi-dimensional exploration: First, conduct gene expression analysis across diverse plant species or within the same species. Based on spatiotemporal expression characteristics of genes, researchers should design targeted stress-resistant breeding strategies. For instance, this study analyzed the expression patterns of foxtail millet RLK-LRR genes in spical, leaves, stems, and roots, and performed homologous comparison analyses with rice genes (Supplementary data). This approach provides a template for leveraging evolutionary conservation in crop improvement programs.”

Comments 17: On the other hand, authors also mentioned a second approach (regulatory networks and integrate information).

Based on that, several R packages for multi-omics purposes are currently available.
Considering the genomic and transcriptomic data analyzed, why do the authors did not integrate this information? And improve the results?

This, will clearly help in the understanding of the mechanism of action mentioned previously.

And therefore, it will not be a perspective.

Response 17: We sincerely thank the reviewer for raising this important point. Regarding the integration of multi-omics data for regulatory network analysis, we would like to clarify that the current study focused on genomic and transcriptomic datasets from a single foxtail millet cultivar (Yugu1). While this preliminary dataset allowed us to identify key LRR-RLK candidates, its limited genetic diversity and sample size currently preclude robust multi-omics integration.

To address this limitation, our team is now systematically sequencing more geographically diverse foxtail millet varieties, including drought-tolerant and salt-resistant landraces. This expanded dataset encompassing pan-genome variation, time-series transcriptomes under abiotic stresses, and miRNA profiles, will provide the necessary foundation for future publication.

Reviewer 2 Report (New Reviewer)

Comments and Suggestions for Authors

Authors have performed the Genome-Wide Identification and Functional Prediction of LRRRLK Family Genes in Foxtail Millet (Setaria italica) in Response to Stress. Foxtail Millet is a vital cereal crop; in this study, the authors have performed a comprehensive phylogenetic analysis and chromosomal mapping of LRR-RLK genes in Setaria italica. Authors are advised to address the following comments to improve the manuscript,

  1. Clarify the specific sequence identity thresholds and e-value cutoffs used in the BLAST analysis to identify LRR-RLK family members in foxtail millet.
  2. How did authors ensure that predicted domains (LRR and kinase) were functional and not just structurally present? Was any additional functional validation (e.g., conserved motif analysis) conducted?
  3. Authors are requested to elaborate on the environmental conditions (e.g., drought intensity, salt concentration) used in transcriptomic datasets to ensure their relevance to foxtail millet natural habitats.
  4. Do authors plan to functionally validate key candidate LRR-RLK genes (e.g., through CRISPR knockout or overexpression lines) to confirm their role in stress tolerance?
  5. Expand the legends for Figures 1 and 2 with more details about the figure.
  6. Figure 3: Add y-axis title

Author Response

Authors have performed the Genome-Wide Identification and Functional Prediction of LRRRLK Family Genes in Foxtail Millet (Setaria italica) in Response to Stress. Foxtail Millet is a vital cereal crop; in this study, the authors have performed a comprehensive phylogenetic analysis and chromosomal mapping of LRR-RLK genes in Setaria italica. Authors are advised to address the following comments to improve the manuscript,

Comments 1: Clarify the specific sequence identity thresholds and e-value cutoffs used in the BLAST analysis to identify LRR-RLK family members in foxtail millet.

Response 1: We thank the reviewer for raising this methodological detail. For the BLAST analysis, a stringent E-value cutoff of <1e-10 was applied to identify LRR-RLK family members in foxtail millet. This threshold was selected based on prior studies of plant RLK families to ensure high-confidence homology while minimizing false positives. The revised sentence reads Line 104 to line 106: “We employed BLAST analysis with the Arabidopsis LRR-RLK protein sequences to identify corresponding sequences in millet (E-value cutoff of <1e-10).”

Comments 2: How did authors ensure that predicted domains (LRR and kinase) were functional and not just structurally present? Was any additional functional validation (e.g., conserved motif analysis) conducted?

Response 2: We appreciate the reviewer’s insightful question regarding functional validation of the predicted LRR and kinase domains. To address this, all identified LRR-RLK members were rigorously screened using SMART databases to confirm the presence of both leucine-rich repeat and kinase domains. In the revised manuscript, we have included Supplementary Data listing the annotation results and detailed gene information.

Comments 3: Authors are requested to elaborate on the environmental conditions (e.g., drought intensity, salt concentration) used in transcriptomic datasets to ensure their relevance to foxtail millet natural habitats.

Response 3: We appreciate the reviewer’s question regarding the environmental conditions in our transcriptomic datasets. In the revised manuscript (lines134 to136), we have added a detailed description of the growth conditions: “Plants were grown under a 16/8-hour light/dark photoperiod, 28°C/22°C day/night temperatures, and 50-60% relative humidity to mimic natural seasonal variations.”

Comments 4: Do authors plan to functionally validate key candidate LRR-RLK genes (e.g., through CRISPR knockout or overexpression lines) to confirm their role in stress tolerance?

Response 4: Thank you for raising this critical question. Based on the functional predictions (Figures 1 and 2) and tissue-specific expression patterns (Figure 5 and Supplementary Data), we plan to prioritize several candidate LRR-RLK genes with potential roles in stress tolerance for further functional validation. CRISPR/Cas9-mediated knockout or overexpression assays will be conducted to confirm their biological significance in stress response mechanisms.

Comments 5: Expand the legends for Figures 1 and 2 with more details about the figure.

Response 5: Thank you for pointing this out. We have added detailed legends to Figure 1 and Figure 2, annotating the meaning of the color coding and branch sizes.

Comments 6: Figure 3: Add y-axis title

Response 6: Thank you for highlighting this issue. We sincerely apologize for the oversight. The missing y-axis label has been added to the revised Figure 3.

Round 2

Reviewer 1 Report (Previous Reviewer 1)

Comments and Suggestions for Authors

The authors perform the changes requested. Article can be accepted for publication.

This manuscript is a resubmission of an earlier submission. The following is a list of the peer review reports and author responses from that submission.

Round 1

Reviewer 1 Report

Comments and Suggestions for Authors

The authors describe that perform a “genome-wide analysis”.

However, there is a lack of bioinformatic methods explaining how the work was done.

Indeed, the perspectives that mention the authors are bioinformatic analysis performed

in this type of manuscript.

On the other hand, the article was classified as “review”.

Further, ithenticate report is 31%, which is too high..

Half of the manuscript is written as research article

and the other half as review. However, several bioinformatic analysis were performed by the authors.

Therefore, it may be a short communication or research article.

I think the authors need to modify the way is written or,

explain in detail manner how the bioinformatic analysis were performed.

Here are my suggestions:

The cites (numbers) in the text should not be as “upper-case”.

According with the journal, it should be “normal-case”…

L68-L73. The authors mentioned model plants or plants well studied.

However, I think the authors should include information

about a close plant to the foxtail millet (cereal crop) in order to highlight the importance of this crop.

L77-L81. The authors should go deeper into the information of this crop.

There is only one paragraph mentioned about this crop. Is this the first genome sequenced?

Is the first transcriptomic? Proteomic?

Authors should include genome or genetics information about this crop to improve the introduction.

Please check if the legend “Methods” is needed. If it is needed, please write it in the manuscript.

L85. Arabidopsis thaliana should be in italics.

Furthermore, the genome of Arabidopsis is deposited in several databases,

please include the online reference (TAIR???)

L87. Please re-write the line, phytozome is a database where several genomes are deposited,

including the millet genome.

L85-L88. The authors described that conducted a search in Arabidopsis

and then performed a BLAST against millet genome. Why do the authors perform this?

Do the millet genome is annotated? If is annotated then, direclty then can identify the sequences.

On the other hand, do the authors perform a BLASTn or BLASp? (please specify this)

L99 and L106 how to the authors constructed the phylogenetic tree?

What language or software was used??? (MEGA?, Phyton??? R???)

what method to they use?? What parameters???

Authors need to include all this information.

L100-L103 How do the authors categorize this? What are the parameters?

What software? Authors should explain this

L130. How do the authors perform the analysis? …

how to they perform the distribution of LRR-RLK family???

L145 and L164. Authors should improve the figures 3 and 5

L145. How to they mapped? What software?

L167. How the transcriptome was performed?

L220. What ERECTA Mean?? Include the meaning.

L318-L322. How these analyses were performed..

Furthermore, the size of letters are small in comparison with the other letters of the manuscript.

L328-L391. All the analyses suggested by the authors should be included in this manuscript.

The authors mentioned bioinformatic that needs to be included in this manuscript

(Network analysis, Pathway analysis, among others)

Reviewer 2 Report

Comments and Suggestions for Authors

The study presents a comprehensive phylogenetic analysis and chromosomal mapping of LRR-RLK genes in Setaria italica. 285 genes were identified and classified into 12 subfamilies based on phylogenetic relationships. Functional analysis of these genes suggested that several SiLRR-RLK family members are involved in key stress response pathways.

General concept comments

The manuscript is presented as a review, unless this type of work could be considered a research activity. Anyway, the topic is relevant and complete. References are adequate and self-references are in low number. Figures and tables are appropriate.

Specific comments 

Scientific names must be always reported in italics.

Figure 4 lacks in definition.

References should be checked according to journal’s rules

Comments on the Quality of English Language

The English could be improved to more clearly express the research.

Reviewer 3 Report

Comments and Suggestions for Authors

In the manuscript “Genome-Wide Identification and Functional Prediction of LRR-RLK Family Genes in Foxtail Millet (Setaria italica) in Response to Stress”, the description mainly corresponds to a research work, it is not a review. The title of the manuscript refers to the fact that it is a research paper. The manuscript shows the authors' research results, with figures and tables with a poor description of some methods (assuming it is a review) and presents review sections.